# Drug-Induced Pulmonary Fibrosis: National Database Analysis

**DOI:** 10.3390/biomedicines12122650

**Published:** 2024-11-21

**Authors:** Olga I. Butranova, Elizaveta N. Terekhina, Sergey K. Zyryanov, Geliia N. Gildeeva, Anna A. Abramova, Yury O. Kustov, Irina L. Asetskaya, Vitaly A. Polivanov

**Affiliations:** 1Department of General and Clinical Pharmacology, Peoples’ Friendship University of Russia Named after Patrice Lumumba (RUDN University), 6 Miklukho-Maklaya St., 117198 Moscow, Russia; 1032173831@rudn.ru (E.N.T.); sergey.k.zyryanov@gmail.com (S.K.Z.); 1142220117@pfur.ru (A.A.A.); 1142240146@pfur.ru (Y.O.K.); asetskaya-il@rudn.ru (I.L.A.); 2Pharmacovigilance Center, Information and Methodological Center for Expert Evaluation, Record and Analysis of Circulation of Medical Products Under the Federal Service for Surveillance in Healthcare, 4-1 Slavyanskaya Square, 109074 Moscow, Russia; pvit74@gmail.com; 3Moscow City Health Department, City Clinical Hospital No. 24, State Budgetary Institution of Healthcare of the City of Moscow, Pistzovaya Str. 10, 127015 Moscow, Russia; 4Department of Organization and Management of the Circulation of Medicines, I.M. Sechenov Federal State Autonomous Educational University of Higher Education—First Moscow State Medical University of the Ministry of Health of the Russian Federation (Sechenov University), 8-2 Trubetskaya St., 119991 Moscow, Russia; gildeeva_g_n@staff.sechenov.ru

**Keywords:** drug-induced pulmonary fibrosis, pharmacovigilance, adverse drug reactions, antineoplastic and immunomodulating agents, rituximab

## Abstract

Background/Objectives: Pulmonary fibrosis (PF) results in a progressive decline of lung function due to scarring. Drugs are among the most common causes of PF. The objective of our study was to reveal the structure of drugs involved in PF development. Methods: we performed a retrospective descriptive pharmacoepidemiologic study on spontaneous reports (SRs) with data on PF registered in the Russian National Pharmacovigilance database for the period from 4 January 2019 to 31 May 2024. Results: A total of 1308 SRs on PF were finally identified with patients mean age of 59.3 ± 23.4 years. Death was reported in 30.7% (*n* = 401) with mean age of 59.9 ± 13.8 years. In the structure of culprit drugs, the following groups were leaders: antineoplastic and immunomodulating agents (51.9%); systemic hormonal preparations, excluding sex hormones and insulins (7.4%); drugs affecting nervous system (7.1%); respiratory system (7.1%); alimentary tract and metabolism (6.5%); and cardiovascular system (5.5%). In the total sample, the top ten drugs were rituximab (5.5%), methotrexate (4.4%), etanercept (4.2%), leflunomide (4.0%), adalimumab (3.7%), tocilizumab (3.3%), abatacept (3.0%), alendronic acid (2.7%), secukinumab (2.6%), and infliximab (2.4%). The number of SRs per year nearly doubled from 2021 to 2022 and from 2022 to 2023 with a maximum peak expected for 2024. Conclusions: Our study demonstrated increased reporting on PF in the National Pharmacovigilance database from 2019 to 2024. We revealed outstanding results for the role of antineoplastic and immunomodulating agents in PF development.

## 1. Introduction

Pulmonary fibrosis (PF) is a chronic pathology involving interstitial lung tissue leading to a gradual decline in respiratory function and negative prognosis [1,2]. Idiopathic PF (IPF) is the most typical form of PF, resulting in progressive damage of the interstitial lung tissue due to an unknown cause. An analysis of 22 publications from 12 countries (for the period from January 2009 to April 2020) revealed the rate of global IPF incidence (per 10,000 of the population). It ranged from 0.35 to 1.30 in the Asia-Pacific region, from 0.75 to 0.93 in North America, and from 0.09 to 0.49 in Europe [3]. A cohort study which included England’s population analyzed the dynamics of IPF incidence and mortality for the period from 2008 to 2018. Mortality dynamics revealed an increase of 53%. Using data from the Clinical Practice Research Datalink Aurum, the authors detected a 25% increase in incidence for its broad version and a 100% increase for the narrow one [4]. EUROSTAT data set analysis revealed a growth of overall IPF mortality rate for European countries in 2013–2018. It increased from 3.70 in 2013 to 4.00 in 2018 (average annual percent change 1.74%, 95% credit interval, CI 0.91–2.59%) [5].

Among the main factors contributing to IPF development, genetic predisposition, environmental factors, and accelerated-aging-associated changes were widely discussed [6]. Drug-induced PF (DIPF) represents a separate problem. The first publication dedicated to the role of drugs in PF development was made in 1964, and peak number of publications in the PubMed Database was detected in 2021 [7]. The increasing role of drugs in lung injuries is witnessed by the results of recent pharmacoepidemiologic studies. Analysis of the U.S. Food and Drug Administration Adverse Event Reporting System (FAERS) indicated a 316% increase in a number of DIPF reports per year [8]. Jo et al. performed a nested case–control study to identify drugs associated with DIPF; the highest odds ratios (ORs) were calculated for epidermal growth factor receptor (EGFR) inhibitors (OR 16.84) and class III antiarrhythmics (OR 7.01) [9]. The Japanese Adverse Drug Event Report database (JADER) disclosed the top 10 culprit drugs: methotrexate, gefitinib, gemcitabine, everolimus, docetaxel, nivolumab, paclitaxel, erlotinib, fluorouracil, and oxaliplatin [10]. Based on FAERS database analysis (from 2000 to 2022 year), the most involved DIPF medicines belonged to antirheumatic (39.4%) and antineoplastic agents (26.4%) [8]. Another study analyzed FAERS from 2004 to 2021 year and revealed antineoplastic, cardiovascular, and antirheumatic agents as the leaders among culprit drugs [11]. Enrichment of pharmaceutical market with new drugs emphasizes the actuality of continuous study of risk associated with DIPF. Published works indicated the growing role of monoclonal antibodies in DIPF [12], revealing association of immune checkpoint inhibitors with lung injury [13], as well as many other anticancer drugs [14].

Assessing National Pharmacovigilance databases can help to accumulate knowledge on DIPF risks in various categories of patients. The quality of knowledge is rising with the increase in quantity of analyzed spontaneous reports (SR) indicating a culprit drug. The aim of our study was to investigate the structure of drugs involved in DIPF as well as patients’ characteristics using the Russian National Pharmacovigilance database, also known as the Automatized Information System “Pharmacovigilance” (AIS).

## 2. Materials and Methods

The central pharmacovigilance organization in the Russian Federation is the Federal Service for Surveillance in Healthcare (Roszdravnadzor), functioning to collect data on adverse events (AEs). The structure and management of AIS “Pharmacovigilance” database complies with the ICH E2B (R3) standard [15]. MedDRA version 25.0 [16] and the built-in WHO algorithm and Naranjo algorithm are used in AIS [17]. We estimated the causal relationship for detected drugs, and only ADRs with a high (certain, probable, and possible) causal relationship (Naranjo algorithm) were included in the final analysis. AEs reporters were mainly healthcare professionals, though pharmaceutical companies’ workers, patients, or their representatives were also involved.

We performed a retrospective descriptive pharmacoepidemiologic study, which included SRs registered in the AIS database in the period from 4 January 2019 to 31 May 2024. Inclusion criteria: “Pulmonary fibrosis” indicated in SRs. We include SRs from Russia and other countries. Duplicates and invalid SRs were excluded. The validity of the SRs was performed according to paragraph 407 of the “Rules of Good Pharmacovigilance Practice of the EAEU” [18], stating the necessity of all four elements in SRs: identifiable reporter; identifiable patient; at least one suspected drug; and at least one suspected ADR. SRs were marked invalid in the absence of any of these four elements. The authors estimated the completeness of information about the suspected drug, adverse drug reaction (ADR), patient’s information (gender, age, weight, diagnosis, etc., but no patient’s personal data), and information about the reporter (there is an automatic confirmation when the report is submitted into the AIS database). To estimate severity of the ADR, paragraph 2 of the “Rules of Good Pharmacovigilance Practice of the EAEU” was used [18].

Figure 1 discloses the process of SRs selection from the AIS database.

Suspected drugs were identified by the International Nonproprietary Names (INN) and structured using principles from the international Anatomical Therapeutic Chemical Classification (ATC).

Statistical analysis was performed. Demographic data were derived from the SRs. Statistical data processing was conducted using the Microsoft Excel 2019 software. Several methods of descriptive statistics were used for all analyzed parameters; qualitative variables were described by absolute (n) and relative (%) values. Mean values; standard deviation; median; interquartile range (IQR); the first quartile, or the middle number between the smallest number and the median of the data set (Q1); and the third quartile, or the middle value between the median and the highest value of the data set (Q3) were determined in our study.

Statistical analysis methods were not used to determine the reliability of the differences in the results obtained since the method of SR analysis does not allow for estimations of the total size of the population.

Next, definitions were used in our study [19]:

“Adverse reaction—A response to a medicinal product, which is noxious and un-intended. Adverse reaction may arise from use of the product within or outside the terms of the marketing authorization or from occupational exposure. Use outside the marketing authorization includes off-label use, overdose, misuse, abuse, and medication errors.

“Causality—In accordance with ICH-E2A, the definition of an adverse reaction im-plies at least a reasonable possibility of a causal relationship between a suspected medicinal product and an adverse event. An adverse reaction, in contrast to an adverse event, is characterized by the fact that a causal relationship between a medicinal product and an occurrence is suspected. For regulatory reporting purposes, as detailed in ICH-E2D, if an event is spontaneously reported, even if the relationship is unknown or unstated, it meets the definition of an adverse reaction. Therefore, all spontaneous reports notified by healthcare professionals or consumers are considered suspected adverse re-actions, since they convey the suspicions of the primary sources, unless the reporters specifically state that they believe the events to be unrelated or that a causal relationship can be excluded”.

“A spontaneous report is an unsolicited communication by a healthcare professional, or consumer to a competent authority, marketing authorisation holder or other organization (e.g., regional pharmacovigilance center, poison control center) that describes one or more suspected adverse reactions in a patient who was given one or more medicinal products. It does not derive from a study or any organized data collection systems”.

## 3. Results

The total number of SRs referred to MedDra high level term “Parenchymal lung disorder” and included in the study was 6907. The final number of SRs on PF after duplicates and invalid SRs exclusion was 1308 (18.9%). The distribution of the number of SRs per year is presented in Figure 2. The dynamics of the number of SRs revealed dramatic growth. The number of reports per year nearly doubled from 2021 to 2022 and from 2022 to 2023. A peak may be expected for 2024 as well since the first 5 months accounted for 74.8% of the total number of SRs in 2023.

Outcome analysis based on SRs data indicated a high proportion of lethal cases (30.3%) and absence of dynamics (27.1%); details are shown in Table 1.

In the sample of SRs with lethal outcome, females accounted for 53% (*n* = 213), males accounted for 44% (*n* = 176), and no data on gender accounted for 3% (*n* = 12). The mean age was 59.9 ± 13.8 years (min = 11, max = 92). By estimating data on anamnesis presented in lethal SRs, we revealed that 23.7% (*n* = 95) had respiratory pathology. The maximum number of lethal SRs was in 2023, though tendency witnessed in 2024 will likely overweight it (Figure 3).

The AIS accumulates SRs from Russia and other countries. We performed analysis of SRs depending on the country of origin. The top ten countries based on the number of SRs are presented in Table 2. The leader was Canada (39.9%, *n* = 522), the USA was second with 19.3% (*n* = 253), and Japan was third with 5.1% (*n* = 66).

The next step was to analyze the dynamics of the number and percentage of SRs per year in the top ten countries. The minimum values of SRs numbers for all countries in the AIS database were in 2019. Most countries revealed two peaks in reporting rates, in 2020 and in 2023, an exception was Argentina with only one peak in 2022 (Figure 4).

The assessment of demographic data in the total sample of SRs revealed male gender in 57.3% (*n* = 750), female gender in 38.0% (*n* = 497), and absence of gender information in 4.7% (*n* = 61). The age distribution of the total sample is presented in Table 3.

The mean age of the total sample was 59.3 ± 23.4 (min = 2 years, max = 102 years); the median was 60 years (Q1 = 43; Q3 = 78; IQR = 35). The absolute majority of SRs included adults and youngest-old people (59.1%). The minimum value of the mean age of patients was detected in Canada (56.3 ± 12.7 years), and the oldest patients were from Argentina (72.1 ± 7.7 years). The analysis of demographic data demonstrated predominance of male gender in the general sample (57.34%), with the maximum seen in Japan (male gender in 80.3%, *n* = 53). In contrast, SRs from Canada included mainly females (84.3%, *n* = 440). Details on demographic data depending on the country of SRs origin are presented in Table 4.

Lethal SRs analysis based on the country of SRs origin demonstrated Canada as a leader (37.9%, *n* = 152 out of 401), the USA was second (20%, *n* = 80), and Japan was third (6.2%, *n* = 25); detailed data are shown in Figure 5.

### 3.1. The Structure of Drugs Involved in PF

The total number of suspected drugs was 6008. The mean number of drugs per one SRs was 4.6 ± 9.1 (min = 1, max = 79). There was only one suspected drug per SR in 58% (*n* = 758), two in 11.2% (*n* = 154), three in 5.4% (*n* = 71), four in 4.1% (*n* = 53), five in 2.5% (*n* = 33), and more than five in 18.3% (*n* = 239).

Among the detected drugs, 0.5% (*n* = 31) had no ATC code and thus were not included in further analysis. The structure of the remaining drugs (*n* = 5977) based on ATC is presented in Table 5. Estimation of the leaders revealed groups L (45.4%), A (10.5%), M (7.7%), R (7.6%), and N (6.5%) as the largest groups.

A causal relationship assessment was performed for each drug. Adverse drug reactions (ADRs) with a high (certain, probable, and possible) causal relationship according to the Naranjo algorithm were detected in 74.4% (*n* = 4452), detailed data are shown in Table 6.

Further analysis included only drugs with a high causal relationship. Reassessment of each ATC group contribution to the general structure emphasized a leading role for group L (51.9%), second was for group H (7.4%), third was for groups N and R (7.1% for each), fourth was for group A (6.5%), and fifth was for group C (5.5%), see Table 7.

Analysis of the contribution of each ATC group in the total structure of high causal relationship drugs involved in PF depending on the country of origin is shown in Figure 6. Among all the top ten countries, ATC group L was the leader in the structure of drugs associated with PF.

#### 3.1.1. Group L Drugs Involved in PF

Group L drugs were identified in 1080 SRs (82.6% of the total number of SRs). A high causal relationship was estimated in 744 SRs (56.9% of the total number of SRs). The total number of suspected group L drugs with a high causal relationship was 2310 with a mean number of 3.1 ± 1.4 per SR. The mean age of patients was 57.4 ± 13.3 years (min = 22; max = 102) and the median was 58 years (Q1 = 44; Q3 = 67; IQR = 23). Females accounted for 85.9% (*n* = 1639), males accounted for 10.5% (*n* = 78), and in 3.6% of SRs (*n* = 27), gender was not indicated. The structure of group L drugs is given in Table 8. Among all drugs from group L, the most involved in PF was subgroup L04A Immunosuppressants, which accounted for up to 66.7% (*n* = 1540). The top five drugs in group L were rituximab (10.6%, *n* = 245), methotrexate (8.4%, *n* = 194), etanercept (8.0%, *n* = 185), leflunomide (7.7%, *n* = 178), and adalimumab (7.1%, *n* = 164).

#### 3.1.2. Group H Drugs Involved in PF

The total number of SRs including drugs from group H was 171 (13.1% of the total SRs). The number of SRs which included group H drugs with a high causal relationship was 158 (12.1% of the total SRs). The number of group H drugs with a high causal relationship amounted to 328 and the mean number of drugs per one SRs was 1.9 ± 0.9. By analyzing SRs data, we revealed that the mean age of patients was 52.7 ± 11.6 years (min = 26, max = 85) and the median was 47.5 years (Q1 = 43; Q3 = 62; IQR = 19). Females accounted for 82.9% (*n* = 131), males accounted for 15.8% (*n* = 25), and in 1.3% of SRs (*n* = 2), gender was not indicated. The structure of group H drugs is demonstrated in Table 9.

A majority, 98.8% (*n* = 324), of all drugs from group H appeared to be from the subgroup H02AB Glucocorticoids. The top five drugs in group H were prednisone (31.7%, *n* = 104), hydrocortisone (19.8%, *n* = 65), dexamethasone (16.2%, *n* = 53), cortisone (10.1%, *n* = 33), and triamcinolone (9.5%, *n* = 31).

#### 3.1.3. Group R Drugs Involved in PF

Group R drugs were detected in 187 SRs (14.3% of the total SRs), and those with a high causality relationship were present in 128 SRs (9.8% of the total SRs). The total number of suspected drugs from group R with a high causality relationship was 316, and the mean number per SR was 2.5 ± 1.3. The mean age of patients was 62.2 ± 13.9 years (min = 7, max = 81) and the median was 65 years (Q1 = 48; Q3 = 72; IQR = 24). Females were 71.1% (*n* = 91), males were 27.3% (*n* = 55), and no gender data were in 1.6% of SRs (*n* = 2). The structure of group R drugs is given in Table 10.

The leading subgroup was R03, a drug for obstructive airway diseases, which accounted for up to 74.7% (*n* = 236). The top five drugs in group R included budesonide (13.0%, *n* = 41), salbutamol (11.1%, *n* = 35), montelukast (10.8%, *n* = 34), pseudoephedrine (10.1%, *n* = 32), and tiotropium bromide (7.0%, *n* = 22).

#### 3.1.4. Group N Drugs Involved in PF

Group N drugs were identified in 160 SRs (12.2% of the total number of SRs), and drugs with a high causal relationship were detected in 143 SRs (10.9% of the total number of SRs). The total number of suspected drugs with a high causal relationship from group N was 314, and the mean number per SR was 2.2 ± 1.1. The patients’ mean age was 55.9 ± 15.9 years (min = 40; max = 96) and the median age was 44 years (Q1 = 43; Q3 = 72; IQR = 29). Female gender was detected in 85.3% (*n* = 122) of SRs, male gender was detected in 14.0% (*n* = 20), and no information on gender was detected in 0.7% (*n* = 1) of SRs. The structure of group N drugs involved in PF is highlighted in Table 11.

A total of 63.1% (*n* = 198) of suspected drugs from group N were from the N02 Analgesics subgroup. The top five drugs in group N were oxycodone and its combinations (25.2%, *n* = 79), pregabalin (13.4%, *n* = 42), morphine (11.8%, *n* = 37), quetiapine (11.1%, *n* = 35), and pramipexole (4.5%, *n* = 14).

#### 3.1.5. Group A Drugs Involved in PF

We revealed drugs from group A in 196 SRs (15.0% of the total number of SRs), and those with a high causal relationship were detected in 152 SRs (11.6% of the total number of SRs). The total number of group A drugs with a high causal relationship was 288. The mean number of drugs per SR was 1.9 ± 0.7. The mean age of patients was 56.3 ± 13.6 years (min = 26, max = 80), the median age was 56 years (Q1 = 44; Q3 = 65; IQR = 21). Females accounted for 79.1% (*n* = 121), males accounted for 19.0% (*n* = 29), and in 1.3% of SRs (*n* = 2), gender was not indicated. Table 12 contains information on the structure of drugs from group A.

A total of 46.9% (*n* = 135) of drugs from group A belonged to subgroup A02, drugs for acid-related disorders. The top five drugs in group A included pantoprazole (30.6%, *n* = 88), sulfasalazine (20.5%, *n* = 59), phthalylsulfathiazole (11.5%, *n* = 33), lansoprazole (8.3%, *n* = 24), and domperidone (7.3%, *n* = 21).

#### 3.1.6. Group C Drugs Involved in PF

Drugs from group C were detected in 142 SRs (10.9% of the total number of SRs). The total number of suspected drugs from group C with a high causal relationship was 245, and the mean number of drugs per one SRs was 1.7 ± 0.9. The mean age of patients indicated in SRs was 66.1 ± 13.1 years (min = 43, max = 90), the median age was 66 years (Q1 = 60.5; Q3 = 78; IQR = 17.5). Females accounted for up to 54.2% (*n* = 77), males accounted for 43.7% (*n* = 62), and in 2.1% of SRs (*n* = 3), gender was not indicated.

Surprisingly, the leader among culprit drugs in this group was Candesartan (18.4%, *n* = 45), Amiodarone was second (13.1%, *n* = 32), and Furosemide was third (11.4%, *n* = 28). The structure of group C drugs is indicated in Table 13.

#### 3.1.7. The Top Ten Drugs Involved in PF

Analysis of the total number of suspected drugs with a high causal relationship allowed us to determine the top ten drugs involved in PF throughout the total sample of SRs (Table 14). Rank one was assigned to rituximab (a chimeric murine/human monoclonal immunoglobulin G1 antibody against CD20), an antineoplastic agent initially developed to treat lymphoma [20]. Among the top ten drugs 90% were from group L, including antineoplastic and immunomodulating agents, and the only one, alendronic acid (2.7, *n* = 122), was from group M’s musculoskeletal system. Half of the top ten drugs were monoclonal antibodies.

## 4. Discussion

Drug-induced interstitial lung injury may form a basis for DIPF development. Among the forms of interstitial lung disease, PF is the most widespread and important, and its development is considered as the final grade of pathological changes in the lung parenchyma [21,22]. Considering acute or subacute forms of drug-induced interstitial lung disease, more than 350 culprit drugs have been identified, and in the case of DIPF, the spectrum was narrowed to >80 drugs [23].

Our research revealed the predominance of group L drugs among drugs involved in PF development. It accounted for up to 45.4% (*n* = 2711) of the total sample of suspected drugs and up to 51.9% (*n* = 2310) among the drugs with estimated high causal relationship. The leadership of antineoplastic and immunomodulating agents among drugs associated with DIPF was demonstrated by a variety of studies. Results of a real-world pharmacovigilance research based on the FAERS database demonstrated that most of the top 20 drugs were from group L, and the contribution of monoclonal antibodies was the greatest, as it was shown for Japan, the USA, and France [11]. Another FAERS database analysis revealed that more than 60% of drugs involved in DIPF were antirheumatic (39.4%) and antineoplastic agents (26.4%) [8]. Identification of safety signals regarding drug-induced interstitial lung injury in thirteen FDA-approved antibody–drug conjugates using FAERS data allowed authors to determine seven drugs with strong signals. They were trastuzumab deruxtecan (reporting odds ratio, ROR = 64.15; 95% CI: from 57.07 to 72.10), enfortumab vedotin (5.24; 3.25–8.43), trastuzumab emtansine (3.62; 2.90–4.53), brentuximab vedotin (3.22; 2.49–4.17), polatuzumab vedotin (2.56; 1.59–4.12), gemtuzumab ozogamicin (2.53; 1.70–3.78), and inotuzumab ozogamicin (2.33; 1.21–4.49) [12]. The predominance of group L drugs was also proved by the results of JADER analysis. Iwasa et al. identified group L as the main group associated with interstitial lung disease which indicates its high involvement in PF [24].

Considering possible mechanisms underlying interstitial lung injury induced by antineoplastic and immunomodulating agents, direct toxic action is suggested as the most possible. Formation of reactive oxygen species (ROS), DNA damage, and the inhibition of new DNA synthesis may result in direct cell injury, which may finally lead to fibrosis [25]. Monoclonal antibodies and other drugs with targeted action may provide an inhibitory effect on the growth of tracheal epithelial cells and decrease repairment of aggravating lung damage [26]. It is known that targeted drugs may contribute to chronic inflammation through the damage of alveolar and bronchial epithelial, finally resulting in fibrosis progression [26].

The first rank drug among the top ten revealed in our study was rituximab. The mechanism of DIPF in the case of this drug is supposed to include damage to cell response, involving mainly T and B cells. Subsequent impairment of mucosal immune response leads to the expansion of pulmonary infections progressing with bronchiectasis and PF [27,28]. Interesting results were demonstrated by Adegunsoye et al. (2023). They performed a nationwide cohort study in the USA to investigate the effect of pre-COVID-19 pharmacotherapy on post-COVID-19 PF development [29]. The highest value of post-COVID-19 PF incidence rate ratio (IRR) was detected for rituximab; it was 2.5 (95% CI: from 1.2 to 5.1). Chemotherapy, in general, resulted in lower IRR value (1.6; 95% CI: from 1.3 to 2.0), and for corticosteroids, IRR was 1.2 (95% CI 1.0 to 1.3). For such a common cause of DIPF, amiodarone accounted for up to 0.8 of the IRR (95% CI: 0.6 to 1.1) [29].

Drug-induced changes in the lung parenchyma were described in detail by published works. For targeted antineoplastic drugs, main findings revealed by computed tomography included ground-glass opacities (70.6%) and air-space consolidations (39.2%), less common were interlobular septal thickening (58.8%), intralobular lines (41.2%), and bronchovascular bundle thickening (19.6%) [30]. High-resolution computed tomography disclosed typical patterns of lung toxicity induced by rituximab. They were isolated ground glass, fibrotic pattern, and alveolar hemorrhage [31].

Methotrexate was the second after rituximab among the top ten drugs involved in PF in our study. It is known to induce changes in lung parenchyma; though in most cases, it is difficult to separate the effect of the drug from the effect of rheumatoid arthritis, a disease, which can lead to interstitial lung disease due to its pathogenesis and is traditionally treated with methotrexate [32,33]. Risk factors of methotrexate-induced pneumonitis included age > 60 years, male gender, pre-existing respiratory pathology, diabetes mellitus, chronic kidney disease, hypoalbuminemia, previous use of disease-modifying antirheumatic drugs, and genetic and environment factors [34]. A systematic review performed in 2021 found 29 articles dedicated to methotrexate-induced fibrotic interstitial lung disease, and thirteen out of them proved a negative role of methotrexate, though gained evidence were of low quality [35]. Possible mechanisms of methotrexate-induced pneumonitis include hypersensitivity, direct drug toxicity, and immunosuppression [36]. It was demonstrated that methotrexate treatment may result in the acceleration of inflammation through the facilitation of monocyte-to-alveolar macrophage differentiation, enhancement of Th17 cell activation, and an increase in the proportion of myofibroblasts [37].

Among anti-TNFα agents, etanercept demonstrated relatively high numbers of reports on lung injury [38], and it was the third rank drug in our analysis of the total sample of SRs on PF. One of the first cases of lung injury due to this drug was published in 2002 [39], and latter studies disclosed variable radiographic phenotypes of etanercept-induced lung injury (CD4+-predominant lymphocytic alveolitis, consistent with a sarcoid-like pattern, and CD8+-predominant pattern, consistent with hypersensitivity pneumonitis-like reaction) [40]. It is considered that the risk of etanercept-induced lung injury is increased in patients with preexisting interstitial lung disease [41]. Etanercept was revealed to cause adverse events such as non-infectious pulmonary fibrosis, non-caseating granuloma, interstitial lung disease, autoimmune disease, and accelerated nodulosis [42]. A systemic review associated risks of accelerated pulmonary rheumatoid nodule formation with patients using etanercept [43].

The fourth drug revealed in our study among the top ten drugs associated with PF was leflunomide, which is mainly used as a disease-modifying antirheumatic drug. A systemic review of forty-one articles disclosing leflunomides association with pulmonary toxicity demonstrated that the worldwide prevalence of leflunomide-induced interstitial pneumonitis was 0.02% with a mortality rate of about 20% [44]. Computed tomography revealed bilateral diffuse parenchymal ground-glass opacities or reticular opacities among the most typical patterns [45,46]. The exact mechanism of interstitial lung injury due to leflunomide is unknown, but in this respect, it is interesting to note the results highlighting mechanism of leflunomide-induced nephrotoxicity. It was demonstrated that leflunomide-induced nephrotoxicity was a result of fibrotic changes developed due to leflunomide-induced TGFβ-stimulated p53/Smad2/3 signaling and glomerular and tubular apoptosis [47]. These results suggest profibrotic activity of leflunomide which may contribute to PF development.

Adalimumab was identified in our study as the fifth among the top ten culprit drugs. It is an anti-TNFα agent which involvement in DIPF was previously described in various clinical cases [38,48,49]. The mechanisms explaining lung interstitial tissue changes induced by adalimumab may include unopposed activity of inflammatory cells resulting in formation of interstitial pneumonitis [48]. This mechanism is important since the standard hypothesis describing PF pathogenesis is based on the activation of various populations of macrophages and lymphocytes, resulting in cytokines and chemokines production enhancement which leads to the activation of fibroblasts and myofibroblasts and the subsequent induction of transforming growth factor beta 1 (TGFβ1) and accumulation of abnormal extracellular matrix [50].

Factors of poor prognosis detected for adalimumab-induced interstitial injury included preexisting interstitial lung diseases, older age, delayed onset of symptoms, and co-administration of other immunosuppressant agents [48].

In recent years, tocilizumab attracted attention as a possible causative agent of interstitial lung injury. On the one hand, there are several clinical cases describing toxic changes in lung parenchyma that were believed to be induced by tocilizumab in different categories of patients (in a 14-year-old girl with polyarticular juvenile idiopathic arthritis [51]; in adults and old patients with rheumatoid arthritis [52,53]; and in adult patients with systemic sclerosis-associated interstitial lung disease [54]). On the other hand, the protective effect of tocilizumab was detected in the experimental model of lipopolysaccharide-induced lung injury [55].

Abatacept is associated with amelioration of interstitial lung damage symptoms [56,57,58,59,60]. In a study including forty-four patients with interstitial lung disease related to rheumatoid arthritis, 18 months of abatacept use resulted in significant worsening in 11.4% of patients [59]. The negative effect of abatacept was illustrated in a clinical case of severe acute respiratory distress syndrome development after abatacept use in a patient with rheumatoid arthritis [60]. The assessment of interstitial lung disease incidence in a cohort of 28,559 patients with rheumatoid arthritis revealed crude incidence ratios (IR) per 1000 people per year. The greatest IR was demonstrated for rituximab (6.15; 95% CI: 4.76 to 7.84), tocilizumab was the second (5.05; 95% CI: 3.47 to 7.12), abatacept was the third (4.46; 95% CI: 3.44 to 5.70), followed by adalimumab (3.43; 95% CI: 2.85 to 4.09), and tofacitinib (1.47; 95% CI: 0.54 to 3.27) [61]. These results suggest the possible involvement of abatacept in lung injury which may finally lead to PF development.

Alendronic acid accounted for up to 2.7% among all drugs revealed as a culprit in our study and was marked as rank eighth among the top ten drugs in the total sample. We did not find any published works disclosing the role of this agent in PF or any interstitial lung injury development. Though there are studies indicating association between the use of alendronate and increased risk of lung cancer, supposing negative functional and structural changes to the lung interstitium [62]. In the FAERS database analysis, alendronate was revealed among the top 50 drugs involved in PF, it had rank 26 [8].

Secukinumab was ranked ninth in our study among drugs involved in PF. It was shown to induce interstitial pneumonia in patients with psoriasis [63,64,65]. Among the risk factors of interstitial lung injury due to secukinumab, age, baseline KL-6 levels, and pre-existing interstitial lung diseases were defined. The mean time for the onset of lung symptoms was 14 months after pharmacotherapy initiation [66].

The last among the top ten drugs involved in PF in our study was infliximab, a well-known cause of lung parenchyma damage [64,67]. Published cases described pulmonary changes in patients with psoriasis [68,69] and in patients with ulcerative colitis [70,71]. Among the 122 reported cases of interstitial lung damage caused by biologic agents, infliximab demonstrated a high rate of involvement [67].

Summarizing published data describing the incidence and possible mechanisms of DIPF development due to anticancer drugs, the negative impact of rituximab is the most obvious and is in line with our results. Precautions should be taken for anti-TNF agents and other biologics, which are mainly mAbs. The increased number of SRs on PF associated with mAbs may be due to the overall increased rates of mAbs consumption in the world. Crescioli et al. (2024) revealed the doubling of the Phase 1 to global approval rate for anticancer mAbs between 2000 and 2009 (14%) and between 2010 and 2019 years (29%) [72]. The share growth of mAbs in the pharmaceutical market in the last 40 years (approximate period of time since first mAbs licensing) is impressive [73]. The expected compound annual growth rate for this group of drugs from 2024 to 2033 is supposed to be 11.07% (237.64 billion USD estimated at the end of 2023; 679.03 billion USD expected by the end of 2033) [74]. The broad involvement of protein kinase inhibitors in PF was demonstrated by the results of the study which included SRs from the JADER (analyzed period from April 2004 to February 2017); these agents accounted for up to 26% among all group L drugs [24].

The second group involved in PF due to our results was group H, systemic hormonal preparations, excluding sex hormones and insulins (*n* = 328, 7.4%). The absolute leaders were corticosteroids. A 23-year analysis of the FAERS database identified corticosteroids as important cause of PF; they were fourth among all culprit pharmacological groups (4.6%) [8]. In the top 50 drugs involved in PF, the authors revealed prednisone to be of rank 11, and prednisolone to be of rank 36 [8]. The role of corticosteroids in PF and interstitial lung disease is discussible. They are well-known agents used to treat interstitial lung disorders, including drug-induced cases [75]. A recent study assessed the effect of corticosteroids use on the outcomes of acute exacerbation of fibrotic interstitial lung disease. The results revealed that corticosteroids were associated with an increased risk of inpatient mortality or transplantation (OR: 4.11; 95% CI: 1.00–16.83; *p* = 0.049), a reduction in the median survival (221 vs. 520.5 days), and an increased risk of all-cause mortality (hazard ratio, HR: 3.25; 95% CI: 1.56–6.77; *p* < 0.01) [76]. A systematic review and meta-analysis of 10 observational studies (*n* = 1639) demonstrated uncertain effects of corticosteroids in patients with fibrotic interstitial lung disease [77].

Groups R and N drugs accounted for up to 7.1% among all drugs with a high causal relationship and shared third place due to our results. Drugs affecting respiratory system may appear in the list of culprit agents due to their broad use in various pathological processes in the lungs involving PF-related cases. Estimation of the role of group R drugs in DIPF requires more randomized clinical trials. Psychoactive drugs are absent among the typical causes of DIPF, though some authors described possible mechanisms of their pulmonary toxicity, taking into account the pathological cascade induced by cocaine [31,78,79].

Group A drugs (alimentary tract and metabolism) ranked fourth among all drugs involved in DIPF according to our data, and pantoprazole was the leader. Published data including systematic reviews demonstrated the protective role of proton pump inhibitors in IPF [80,81]. The FAERS database assessment by Jiang et al. (2024) revealed one of the proton pump inhibitors (esomeprazole) among the list of top drugs involved in interstitial lung disease in France (1.68%, *n* = 64) [11]. Sulfasalazine was the second among group A drugs involved in DIPF in our study. The phenomenon of sulfasalazine-induced pulmonary toxicity has been known since 1972 [82], and modern studies proved its involvement in multiple lung tissue disorders [83].

The last group among top five pharmacological groups involved in DIPF due to our results was group C. Traditionally, amiodaron is the most known cause of DIPF among cardiovascular agents, but we revealed a higher frequency for candesartan, an angiotensin converting enzyme (ACE)-inhibitor. Considering the role of candesartan in lung injury, it is important to note the results of other studies which witnessed its beneficial effects and ability to attenuate pathological changes in the lung tissue induced by other drugs, mainly anticancer agents [84,85]. ACE-inhibitors were able to mitigate radiation-induced pneumonitis through inhibition of ACE expression and decrease in ROS formation [86]. The appearance of candesartan in the list of drugs involved in DIPF demands additional investigations and relevant data derived from randomized clinical trials.

A discussion of the results would be incomplete without discussing the limitations of our study. First, we need to emphasize that retrospective analysis of SRs entered into the pharmacovigilance database allowed us to identify the list of culprit drugs but not the size of the total sample of population using these specific drugs. Therefore, the incidence of DIPF related to a particular pharmacologic group cannot be calculated. Despite the presence of demographic data, concomitant diseases, and list of medications in SRs, we did not have enough information to investigate the risk factors of PF development. Moreover, it should be stated that the diagnosis of PF is a complex process, which requires a multidisciplinary approach, integrating experts from different fields of medicine [87], and the number of SRs in the AIS database does not reflect the overall rate of DIPF prevalence in the population. Further studies are needed to elucidate the mechanisms of lung injury leading to DIPF development, including genetic studies, which may help to determine individual risk factors. The value of estimation of genetic factors may be accentuated by the results of the study performed by Drent et al. (2024), which revealed that in the cohort of patients with suspected drug-induced lung injury 79%, patients carried one or more genetic variants accompanied by the use of drugs metabolized by a corresponding affected pathway [88].

## 5. Conclusions

Our study emphasized the actuality of DIPF problems, demonstrating a dramatic increase in the number of SRs in the National Pharmacovigilance database from 2019 to 2024. We revealed outstanding results for the group with antineoplastic and immunomodulating agents in PF development; they accounted for about half of all drugs with a high causal relationship (51.9%) with most of the cases associated with rituximab. The contribution of other pharmacological groups identified in our study was significantly smaller. In second place with 7.4% was group H, third place with 7.1% for both groups N and R, fourth place with 6.5% for group A, and fifth place with only 5.5% for group C. In the structure of the top ten drugs involved in PF, nine were from group L, and only one from group M (alendronic acid). The demonstrated results proved the involvement of mAbs in PF development, where half of the top ten drugs were mAbs.

## Figures and Tables

**Figure 1 biomedicines-12-02650-f001:**
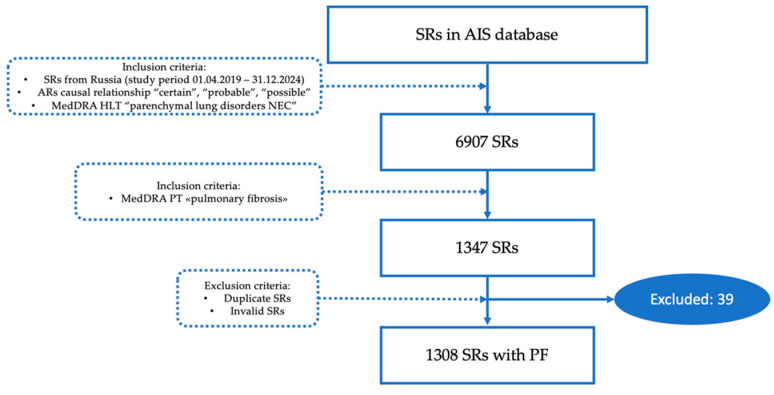
Flowchart of SRs selection from the AIS “Pharmacovigilance” (AR—adverse reaction; HLT—high-level term; PT—preferred term; SR—spontaneous report).

**Figure 2 biomedicines-12-02650-f002:**
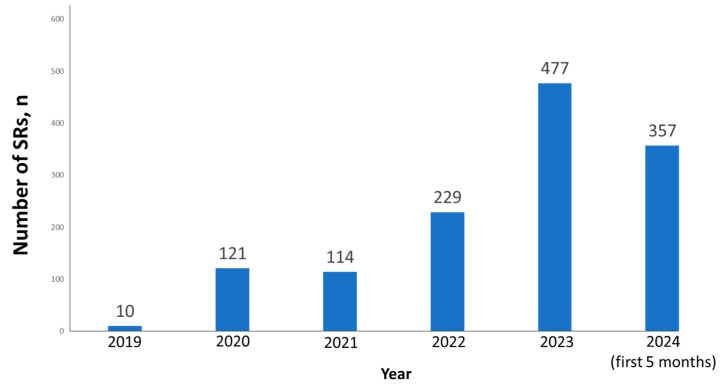
The distribution of the number of SRs per year.

**Figure 3 biomedicines-12-02650-f003:**
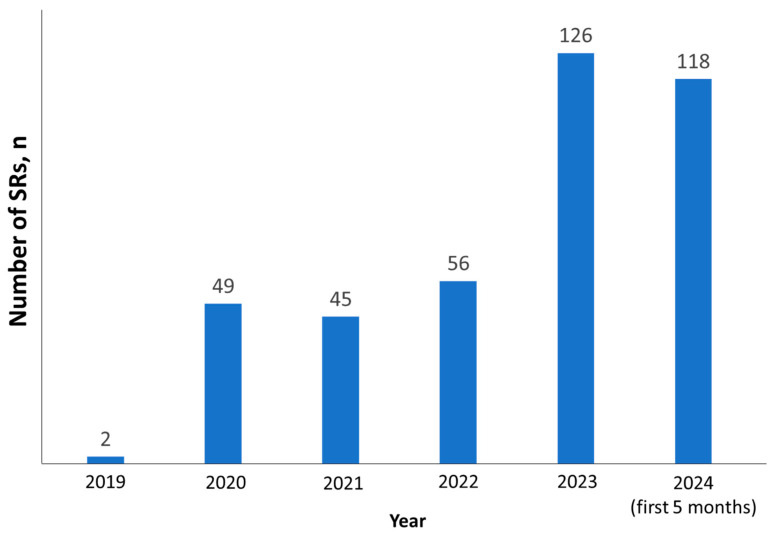
Dynamics of the number of lethal SRs.

**Figure 4 biomedicines-12-02650-f004:**
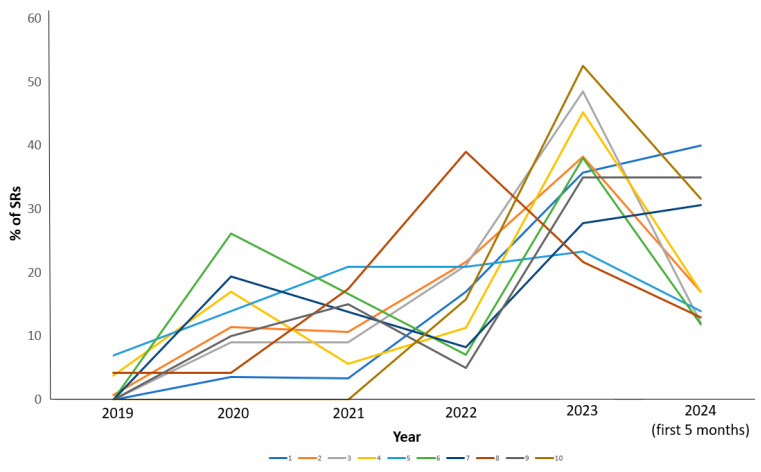
Dynamics of % SRs per year in the top 10 countries (1—Canada, 2—USA, 3—Japan, 4—Germany, 5—Russia, 6—France, 7—Brazil, 8—Argentina, 9—United Kingdom, and 10—China).

**Figure 5 biomedicines-12-02650-f005:**
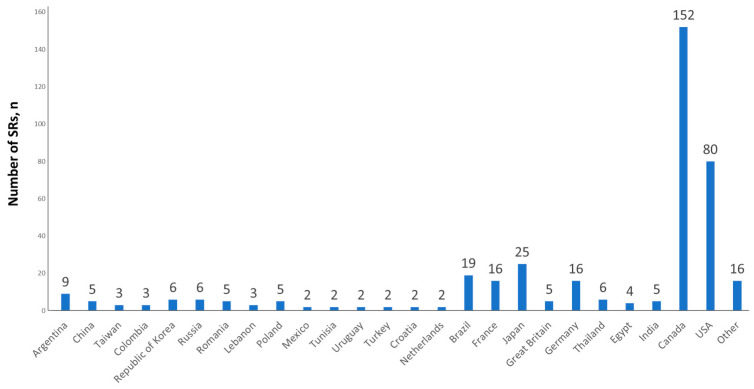
Number of lethal SRs in different countries detected in the AIS database.

**Figure 6 biomedicines-12-02650-f006:**
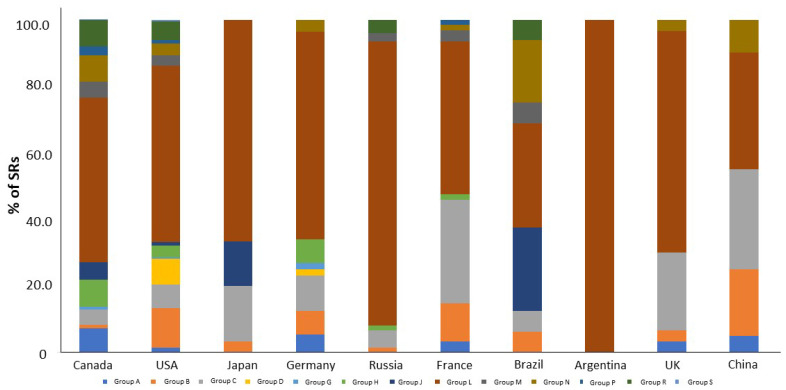
Contribution of each ATC group in the total structure of high causal relationship drugs involved in PF depending on country.

**Table 1 biomedicines-12-02650-t001:** SRs on PF: outcome analysis.

Outcome	N (Total—1308)	%
Death	401	30.7
Condition unchanged	355	27.1
Condition improved	64	4.0
Recovery with consequences	8	0.6
Recovery without consequences	94	7.2
Unknown	386	29.5

**Table 2 biomedicines-12-02650-t002:** SRs distribution by country of origin.

Country	N (Total—1308)	%
Canada	522	39.9
USA	253	19.3
Japan	66	5.1
Germany	53	4.1
Russia	43	3.3
France	42	3.2
Brazil	36	2.8
Argentina	23	1.8
UK	20	1.5
China	19	1.5
Other	231	17.7

**Table 3 biomedicines-12-02650-t003:** Age Distribution of Patients with PF.

Age Group	N (Total—1308)	%
0–18 (children)	3	0.2
19–59 years (adults)	370	28.3
60–74 years (youngest-old)	403	30.8
75–89 years (middle-old)	202	15.4
≥85 years (oldest-old)	11	0.8
No data	319	24.4

**Table 4 biomedicines-12-02650-t004:** Demographic Data Based on SRs from the Top Ten Countries.

Country	Gender						Age		
	Female		Male		Unknown		Mean (±SD)	Min	Max
	*n*	%	*n*	%	*n*	%			
Canada	440	84.3	74	14.2	8	1.5	56.3 ± 12.7	28	89
USA	114	45.1	114	45.1	25	9.9	71.0 ± 10.8	33	96
Japan	10	15.2	53	80.3	3	4.5	71.3 ± 7.7	43	87
Germany	22	41.5	29	54.7	2	3.8	59.5 ± 16.8	26	87
Russia	21	48.8	20	46.5	2	4.7	56.4 ± 15.1	7	87
France	15	35.7	24	57.1	3	7.1	71.2 ± 9.8	42	87
Brazil	19	52.8	17	47.2	0	0	68.8 ± 17.5	2	90
Argentina	12	52.2	10	43.5	1	4.3	72.1 ± 7.7	57	83
UK	10	50.0	8	40.0	2	10.0	67.4 ± 19.6	42	88
China	9	47.4	8	42.1	2	10.5	61.7 ± 18.7	23	98

**Table 5 biomedicines-12-02650-t005:** The Structure of Drugs Involved in PF (ATC groups).

ATC Group	N (Total—5977)	%
A—Alimentary Tract and Metabolism	625	10.50
B—Blood and Blood forming organs	179	3.0
C—Cardiovascular System	249	4.20
D—Dermatologicals	145	2.40
G—Genito Urinary System and Sex Hormones	41	0.70
H—Systemic Hormonal Preparations, excl. Sex Hormones and Insulins	352	5.90
J—Antiinfectives for Systemic Use	251	4.20
L—Antineoplastic and Immunomodulating Agents	2711	45.40
M—Musculo-Skeletal System	461	7.70
N—Nervous System	387	6.50
P—Antiparasitic Products, Insecticides and Repellents	109	1.80
R—Respiratory System	457	7.60
S—Sensory Organs	3	0.05
V—Various	7	0.10

**Table 6 biomedicines-12-02650-t006:** Results of Causal Relationship Assessment Among Suspected Drugs.

Causal Relationship	N (Total—5977)	%
Certain	1131	18.9
Probable	1018	17.0
Possible	2303	38.5
Unlikely	1084	18.1
Unassessable	441	7.4

**Table 7 biomedicines-12-02650-t007:** The Structure of Drugs with High Causal Relationship Involved in PF.

ATC Group	N (Total—4452)	%
A—Alimentary Tract and Metabolism	288	6.5
B—Blood and Blood forming organs	84	1.9
C—Cardiovascular System	245	5.5
D—Dermatologicals	18	0.4
G—Genito Urinary System and Sex Hormones	34	0.8
H—Systemic Hormonal Preparations, excl. Sex Hormones and Insulins	328	7.4
J—Antiinfectives for Systemic Use	208	4.7
L—Antineoplastic and Immunomodulating Agents	2310	51.9
M—Musculo-Skeletal System	198	4.4
N—Nervous System	314	7.1
P—Antiparasitic Products, Insecticides and Repellents	106	2.4
R—Respiratory System	316	7.1
S—Sensory Organs	3	0.1

**Table 8 biomedicines-12-02650-t008:** The Structure of Group L Drugs Involved in PF.

Drugs	N (Total—2310)	%
L01 Antineoplastic Agents	747	32.3
L01A Alkylating Agents	18	0.8
L01AA Nitrogen Mustard Analogs	14	0.6
Cyclophosphamide	9	0.4
Bendamustine	5	0.2
L01AB Alkyl Sulfonates	2	0.1
Busulfan	2	0.1
L01AD Nitrosoureas	1	0
Lomustine	1	0
L01AX Other Alkylating Agents	1	0
Dacarbazine	1	0
L01B Antimetabolites	224	9.7
L01BA Folic Acid Analogs	210	9.1
Methotrexate	194	8.4
Pemetrexed	16	0.7
L01BB Purine Analogs	1	0
Mercaptopurine	1	0
L01BC Pyrimidine Analogs	13	0.6
Cytarabine	2	0.1
Fluorouracil	2	0.1
Gemcitabine	4	0.2
Capecitabine	2	0.1
Azacitidine	3	0.1
L01C Plant Alkaloids and Other Natural Products	29	1.3
L01CA Vinca Alkaloids and Analogs	6	0.3
Vinblastine	1	0
Vincristine	3	0.1
Vinorelbine	2	0.1
L01CB Podophyllotoxin Derivatives	2	0.1
Etoposide	2	0.1
L01CD Taxanes	21	0.9
Paclitaxel	9	0.3
Docetaxel	12	0.5
L01D Cytotoxic Antibiotics and Related Substances	11	0.5
L01DB Anthracyclines and Related Substances	5	0.2
Doxorubicin	5	0.2
L01DC Other Cytotoxic Antibiotics	6	0.3
Bleomycin	6	0,3
L01E Protein Kinase Inhibitors	83	3.6
L01EA BCR-ABL Tyrosine Kinase Inhibitors	7	0.3
Imatinib	4	0.2
Dasatinib	1	0
Bosutinib	1	0
Asciminib	1	0
L01EB Epidermal Growth Factor Receptor (EGFR) Tyrosine Kinase Inhibitors	6	0.3
Afatinib	3	0.1
Dacomitinib	3	0.1
L01EC B-Raf Serine-Threonine Kinase (BRAF) Inhibitors	3	0.1
Vemurafenib	1	0
Encorafenib	2	0.1
L01ED Anaplastic Lymphoma Kinase (ALK) Inhibitors	12	0.5
Crizotinib	8	0.3
Alectinib	1	0
Lorlatinib	3	0.1
L01EE Mitogen-Activated Protein Kinase (MEK) Inhibitors	3	0.1
Cobimetinib	1	0
Binimetinib	2	0.1
L01EF Cyclin-Dependent Kinase (CDK) Inhibitors	35	1.5
Palbociclib	23	1.0
Ribociclib	3	0.1
Abemaciclib	9	0.4
L01EG Mammalian Target of Rapamycin (mTOR) Kinase Inhibitors	2	0.1
Everolimus	2	0.1
L01EH Human Epidermal Growth Factor Receptor 2 (HER2) Tyrosine Kinase Inhibitors	1	0
Lapatinib	1	0
L01EJ Janus-Associated Kinase (JAK) Inhibitors	3	0.1
Ruxolitinib	3	0.1
L01EK Vascular Endothelial Growth Factor Receptor (VEGFR) Tyrosine Kinase Inhibitors	1	0
Axitinib	1	0
L01EL Bruton’s Tyrosine Kinase (BTK) Inhibitors	4	0.2
Ibrutinib	4	0.2
L01EM Phosphatidylinositol-3-Kinase (Pi3K) Inhibitors	1	0
Duvelisib	1	0
L01EX Other Protein Kinase Inhibitors	5	0.2
Lenvatinib	1	0
Gilteritinib	3	0.1
Capmatinib	1	0
L01F Monoclonal Antibodies and Antibody Drug Conjugates	344	14.9
L01FA CD20 (Clusters of Differentiation 20) Inhibitors	247	10.7
Rituximab	245	10.6
Obinutuzumab	2	0.1
L01FD HER2 (Human Epidermal Growth Factor Receptor 2) Inhibitors	21	0.9
Trastuzumab	10	0.4
Pertuzumab	5	0.2
Trastuzumab emtansine	6	0.3
L01FE EGFR (Epidermal Growth Factor Receptor) Inhibitors	3	0.1
Cetuximab	1	0
Panitumumab	2	0.1
L01FF PD-1/PD-L1 (Programmed Cell Death Protein 1/Death Ligand 1) Inhibitors	49	2.1
Nivolumab	12	0.5
Pembrolizumab	22	1.0
Atezolizumab	14	0.6
Dostarlimab	1	0,
L01FG VEGF/VEGFR (Vascular Endothelial Growth Factor) Inhibitors	11	0.5
Bevacizumab	11	0.5
L01FX Other Monoclonal Antibodies and Antibody Drug Conjugates	13	0.6
Ipilimumab	3	0.1
Brentuximab Vedotin	4	0.2
Blinatumomab	1	0
Elotuzumab	1	0
Enfortumab Vedotin	2	0.1
Belantamab	1	0
Teclistamab	1	0
L01X Other Antineoplastic Agents	38	1.6
L01XA Platinum Compounds	33	1.4
Cisplatin	5	0.2
Carboplatin	19	0.8
Oxaliplatin	9	0.4
L01XG Proteasome Inhibitors	2	0.1
Bortezomib	2	0.1
L01XK Poly (ADP-ribose) Polymerase (PARP) Inhibitors	1	0
Rucaparib	1	0
L01XX Other Antineoplastic Agents	2	0.1
Hydroxycarbamide	1	0
Sotorasib	1	0
L02 Endocrine Therapy	19	0.8
L02A Hormones and Related Agents	1	0
L02AE Gonadotropin Releasing Hormone Analogs	1	0
Triptorelin	1	0
L02B Hormone Antagonist and Related Agents	18	0.8
L02BA Anti-Estrogens	5	0.2
Tamoxifen	3	0.1
Fulvestrant	2	0.1
L02BB Anti-Androgens	10	0.4
Enzalutamide	4	0.2
Apalutamide	5	0.2
Darolutamide	1	0
L02BG Aromatase Inhibitors	2	0.1
Letrozole	1	0
Exemestane	1	0
L02BX Other Hormone Antagonists and Related Agents	1	0
Abiraterone	1	0
L03 Immunostimulants	4	0.2
L03A Immunostimulants	4	0.2
L03AA Colony Stimulating factors	1	0
Filgrastim	1	0
L03AB Interferons	2	0.1
Pegylated Interferon Alfa-2a	2	0.1
L03AX Other Immunostimulants	1	0
BCG Vaccine	1	0
L04 Immunosuppressants	1540	66.7
L04A Immunosuppressants	1540	66.7
L04AA Selective Immunosuppressants	262	11.3
Antithymocyte Immunoglobulin	1	0
Mycophenolic Acid	78	3.4
Abatacept	132	5.7
Apremilast	51	2.2
L04AB Tumor Necrosis Factor Alpha (TNF-α) Inhibitors	579	25.1
Etanercept	185	8.0
Infliximab	107	4.6
Adalimumab	164	7.1
Certolizumab Pegol	60	2.6
Golimumab	63	2.7
L04AC Interleukin Inhibitors	326	14.1
Anakinra	14	0.6
Ustekinumab	43	1.9
Tocilizumab	148	6.4
Secukinumab	116	5.0
Ixekizumab	3	0.1
Risankizumab	1	0
Satralizumab	1	0
L04AE Sphingosine-1-Phosphate (S1P) Receptor Modulators	2	0.1
Fingolimod	1	0
Siponimod	1	0
L04AF Janus-Associated Kinase (JAK) Inhibitors	105	4.5
Tofacitinib	105	4.5
L04AG Monoclonal Antibodies	11	0.5
Vedolizumab	5	0.2
Alemtuzumab	2	0.1
Ocrelizumab	3	0.1
Ofatumumab	1	0
L04AJ Complement Inhibitors	1	0
Avacopan	1	0
L04AK Dihydroorotate Dehydrogenase (DHODH) Inhibitors	181	7.8
Leflunomide	178	7.7
Teriflunomide	3	0.1
L04AX Other Immunosuppressants	73	3.2
Azathioprine	51	2.2
Lenalidomide	15	0.6
Pomalidomide	7	0.3

**Table 9 biomedicines-12-02650-t009:** The Structure of Group H Drugs Involved in PF.

Drugs	N (Total—328)	%
H02 Corticosteroids for Systemic Use	324	98.8%
H02A Corticosteroids for Systemic Use, Plain	324	98.8%
H02AB Glucocorticoids	324	98.8%
Betamethasone	7	2.1%
Dexamethasone	53	16.2%
Methylprednisolone	17	5.2%
Prednisolone	14	4.3%
Prednisone	104	31.7%
Triamcinolone	31	9.5%
Hydrocortisone	65	19.8%
Cortisone	33	10.1%
H05 Calcium Homeostasis	4	1.2%
H05A Parathyroid Hormones and Analogs	4	1.2%
H05AA Parathyroid Hormones and Analogs	4	1.2%
Teriparatide	4	1.2%

**Table 10 biomedicines-12-02650-t010:** The Structure of Group R Drugs Involved in PF.

Drugs	N (Total—316)	%
R01 Nasal Preparations	74	23.4
R01A Decongestants and Other Nasal Preparations for Topical Use	22	7.0
R01AA Sympathomimetics, Plain	19	6.0
Phenylephrine	17	5.4
Xylometazoline	2	0.6
R01AD Corticosteroids	2	0.6
Fluticasone Furoate	2	0.6
R01AX Other Nasal Preparations	1	0.3
Framycetin	1	0.3
R01B Nasal Decongestants for Systemic Use	52	16.5
R01BA Sympathomimetics	52	16.5
Pseudoephedrine	32	10.1
Pseudoephedrine Hydrochloride + Cetirizine Dihydrochloride	3	0.9
Cetirizine + Pseudoephedrine	17	5.4
R02 Throat Preparations	6	1.9
R02AB Antibiotics	6	1.9
Tyrothricin	2	0.6
Gramicidin	4	1.3
R03 Drugs for Obstructive Airway Diseases	236	74.7
R03A Adrenergics, Inhalants	76	24.1
R03AC Selective Beta-2-Adrenoreceptor Agonists	45	14.2
Salbutamol	35	11.1
Salmeterol	2	0.6
Formoterol	8	2.5
R03AK Adrenergics in Combination with Corticosteroids or Other Drugs, excl. Anticholinergics	17	5.4
Salmeterol and Fluticasone	3	0.9
Budesonide + Formtorol	8	2.5
Formoterol + Beclometasone	1	0.3
Formoterol and Mometasone	2	0.6
Vilanterol + Fluticasone Furoate	3	0.9
R03AL Adrenergics in Combination with Anticholinergics incl. Triple Combinations with Corticosteroids	14	4.4
Fenoterol + Ipratropium	1	0.3
Ipratropium Bromide + Salbutamol	1	0.3
Vilanterol + Umeclidinium Bromide	7	2.2
Aclidinium Bromide + Formoterol	1	0.3
Olodaterol + Tiotropium Bromide	3	0.9
Vilanterol	1	0,3
R03B Other Drugs for Obstructive Airway Diseases, Inhalants	101	32.0
R03BA Glucocorticoids	60	19.0
Budesonide	41	13.0
Fluticasone	3	0.9
Mometasone	13	4.1
Ciclesonide	3	0.9
R03BB Anticholinergics	41	13.0
Ipratropium Bromide	12	3.8
Tiotropium Bromide	22	7.0
Aclidinium Bromide	1	0.3
Umeclidinium Bromide	6	1.9
R03D Other Systemic Drugs for Obstructive Airway Diseases	59	18.7
R03DC Leukotriene Receptor Antagonists	34	10.8
Montelukast	34	10.8
R03DX Other Systemic Drugs for Obstructive Airway Diseases	25	7.9
Omalizumab	13	4.1
Mepolizumab	12	3.8

**Table 11 biomedicines-12-02650-t011:** The Structure of Group N Drugs Involved in PF.

Drugs	N (Total—314)	%
N02 Analgesics	198	63.1
N02A Opioids	141	44.9
N02AA Natural Opium Alkaloids	101	32.2
Morphine	37	11.8
Hydromorphone	2	0.6
Oxycodone	62	19.7
N02AE Oripavine Derivatives	1	0.3
Buprenorphine	1	0.3
N02AJ Opioids in Combination with Non-Opioid Analgesics	28	8.9
Codeine + Paracetamol	5	1.6
Oxycodone + Paracetamol	17	5.4
Caffeine + Codeine + Paracetamol	4	1.3
Paracetamol + Tramadol + Vitamin B1 + Vitamin B12	2	0.6
N02AX Other Opioids	11	3.5
Tramadol	11	3.5
N02B Other Analgesics and Antipyretics	46	14.6
N02BB Pyrazolones	2	0.6
Metamizole Sodium	2	0.6
N02BF Gabapentinoids	44	14.0
Gabapentin	2	0.6
Pregabalin	42	13.4
N02C Antimigraine Preparations	11	3.5
N02CD Calcitonin Gene-Related Peptide (CGRP) Antagonists	11	3.5
Erenumab	11	3.5
N03 Antiepileptics	7	2.2
N03A Antiepileptics	7	2.2
N03AE Benzodiazepine Derivatives	1	0.3
Clonazepam	1	0.3
N03AG Fatty Acid Derivatives	2	0.6
Valproic Acid	2	0.6
N03AX Other Antiepileptics	4	1.3
Lamotrigine	1	0.3
Topiramate	2	0.6
Levetiracetam	1	0.3
N04 Anti-Parkinson Drugs	15	4.8
N04B Dopaminergic Agents	15	4.8
N04BA Dopa and Dopa Derivatives	1	0.3
Levodopa + Benserazide	1	0.3
N04BC Dopamine Agonists	14	4.5
Pramipexole	14	4.5
N05 Psycholeptics	51	16.2
N05A Antipsychotics	35	11.1
N05AH Diazepines, Oxazepines, Thiazepines and Oxepines	35	11.1
Quetiapine	35	11.1
N05B Anxiolytics	2	0.6
N05BA Benzodiazepine Derivatives	2	0.6
Lorazepam	2	0.6
N05C Hypnotics and Sedatives	14	4.5
N05CF Benzodiazepine Related Drugs	14	4.5
Zopiclone	14	4.5
N06 Psychoanaleptics	33	10.5
N06A Antidepressants	31	9.9
N06AA Non-selective Monoamine Reuptake Inhibitors	2	0.6
Opipramol	1	0.3
Amitriptyline	1	0.3
N06AB Selective Serotonin Reuptake Inhibitors	18	5.7
Fluoxetine	2	0.6
Citalopram	1	0.3
Paroxetine	1	0.3
Sertraline	9	2.9
Escitalopram	5	1.6
N06AX Other Antidepressants	11	3.5
Trazodone	8	2.5
Bupropion	1	0.3
Duloxetine	1	0.3
Agomelatine	1	0.3
N06B Psychostimulants, Agents Used for ADHD and Nootropics	2	0.6
N06BA Centrally Acting Sympathomimetics	2	0.6
Atomoxetine	2	0.6
N07 Other Nervous System Drugs	10	3.2
N07A Parasympathomimetics	1	0.3
N07AA Anticholinesterases	1	0.3
Distigmine	1	0.3
N07B Drugs Used in Addictive Disorders	1	0.3
N07BC Drugs Used in Opioid Dependence	1	0.3
Methadone	1	0.3
N07X Other Nervous System Drugs	8	2.5
N07XX Other Nervous System Drugs	8	2.5
Tafamidis	6	1.9
Triprolidine Hydrochloride + Pseudoephedrine Hydrochloride + Dextromethorphan Hydrobromide	1	0.3
Dextromethorphan Hydrobromide + Pseudoephedrine Hydrochloride + Triprolidine Hydrochloride	1	0.3

**Table 12 biomedicines-12-02650-t012:** The Structure of Group A Drugs Involved in PF.

Drugs	N (Total—288)	%
A02 Drugs for Acid Related Disorders	135	46.9
A02B Drugs for Peptic Ulcer and Gastro-Oesophageal Reflux Disease (GORD)	135	46.9
A02BA H2-Receptor Antagonists	14	4.9
Cimetidine	1	0.3
Ranitidine	13	4.5
A02BC Proton Pump Inhibitors	121	42.0
Omeprazole	1	0.3
Pantoprazole	88	30.6
Lansoprazole	24	8.3
Dexlansoprazole	7	2.4
Calcium Stearate + Pantoprazole + Purified Water	1	0.3
A03 Drugs for Functional Gastrointestinal Disorders	21	7.3
A03F Propulsives	21	7.3
A03FA Propulsives	21	7.3
Domperidone	21	7.3
A04 Antiemetics and Antinauseants	15	5.2
A04AA Serotonin (5HT3) Antagonists	1	0.3
Ondansetron	1	0.3
A04AD Other Antiemetics	14	4.9
Aprepitant	12	4.2
Fosaprepitant	2	0.7
A07 Antidiarrheals, Intestinal Antiinflammatory/Antiinfective Agents	109	37.8
A07A Intestinal Antiinfectives	35	12.2
A07AA Antibiotics	2	0.7
Nystatin	2	0.7
A07AB Sulfonamides	33	11.5
Phthalylsulfathiazole	33	11.5
A07E Intestinal Antiinflammatory Agents	74	25.7
A07EC Aminosalicylic Acid and Similar Agents	74	25.7
Sulfasalazine	59	20.5
Mesalazine	15	5.2
A10 Drugs Used in Diabetes	5	1.7
A10B Blood Glucose Lowering Drugs, excl. Insulins	5	1.7
A10BB Sulfonylureas	5	1.7
Glibenclamide	1	0.3
Gliclazide	3	1.0
Glimepiride	1	0.3
A12 Mineral Supplements	2	0.7
A12C Other Mineral Supplements	2	0.7
A12CD Fluoride	2	0.7
Sodium Fluoride	2	0.7
A16 Other Alimentary Tract and Metabolism Products	1	0.3
A16A Other Alimentary Tract and Metabolism Products	1	0.3
A16AB Enzymes	1	0.3
Agalsidase Beta	1	0.3

**Table 13 biomedicines-12-02650-t013:** Group C Drugs Involved in PF.

Drug	N (Total—245)	%
C01 Cardiac Therapy	46	18.8
C01B Antiarrhythmics, Class I and III	33	13.5
C01BD Antiarrhythmics, Class III	33	13.5
Amiodarone	32	13.1
Dronedaron	1	0.4
C01D Vasodilators Used in Cardiac Diseases	13	5.3
C01DA Organic Nitrates	13	5.3
Glyceryl Trinitrate	12	4.9
Isosorbide Mononitrate	1	0.4
C02 Antihypertensives	13	5.3
C02A Antiadrenergic Agents, Centrally Acting	7	2.9
C02AC Imidazoline Receptor Agonists	7	2.9
Clonidine	7	2.9
C02C Antiadrenergic Agents, Peripherally Acting	4	1.6
C02CA Alpha-Adrenoreceptor Antagonists	4	1.6
Doxazosin	4	1.6
C02K Other Antihypertensives	2	0.8
C02KX Antihypertensives for Pulmonary Arterial Hypertension	2	0.8
Macitentan	2	0.8
C03 Diuretics	50	20.4
C03A Low-ceiling Diuretics, Thiazides	19	7.8
C03AA Thiazides, Plain	18	7.3
Hydrochlorothiazide	17	6.9
Trichlormethiazide	1	0.4
C03AX Thiazides, Combinations with Other Drugs	1	0.4
Hydrochlorothiazide + Irbesartan	1	0.4
C03B Low-ceiling Diuretics, excl. Thiazides	1	0.4
C03BA Sulfonamides, Plain	1	0.4
Indapamide	1	0.4
C03C High-ceiling Diuretics	28	11.4
C03CA Sulfonamides, Plain	28	11.4
Furosemide	28	11.4
C03D Aldosterone Antagonists and Other Potassium-Sparing Agents	2	0.8
C03DA Aldosterone Antagonists	2	0.8
Spironolactone	2	0.8
C07 Beta Blocking Agents	12	4.9
C07A Beta Blocking Agents	12	4.9
C07AB Beta Blocking Bgents, Selective	12	4.9
Metoprolol	4	1.6
Bisoprolol	6	2.4
Celiprol	1	0.4
Nebivolol	1	0.4
C08 Calcium Channel Blockers	18	7.3
C08C Selective Calcium Channel Blockers with Mainly Vascular Effects	4	1.6
C08CA Dihydropyridine Derivatives	4	1.6
Amlodipine	3	1.2
Benidipin	1	0.4
C08D Selective Calcium Channel Blockers with Direct Cardiac Effects	14	5.7
C08DB Benzothiazepine Derivatives	14	5.7
Diltiazem	14	5.7
C09 Agents Acting on the Renin-Angiotensin System	80	32.7
C09A ACE Inhibitors, Plain	28	11.4
C09AA ACE Inhibitors, Plain	28	11.4
Lisinopril	7	2.9
Ramipril	21	8.6
C09C Angiotensin II Receptor Blockers (ARBs), Plain	51	20.8
C09CA Angiotensin II Receptor Blockers (ARBs), Plain	51	20.8
Losartan	1	0.4
Valsartan	3	1.2
Candesartan	45	18.4
Telmisartan	2	0.8
C09D Angiotensin II Receptor Blockers (ARBs), Combinations	1	0.4
C09DX Angiotensin II Receptor Blockers (ARBs), Other Combinations	1	0.4
Sacubitril + Valsartan	1	0.4
C10 Lipid Modifying Agents	26	10.6
C10A Lipid Modifying Agents, Plain	26	10.6
C10AA HMG CoA Reductase Inhibitors	25	10.2
Simvastatin	2	0.8
Lovastatin	1	0.4
Atorvastatin	3	1.2
Rosuvastatin	19	7.8
C10AX Other Lipid Modifying Agents	1	0.4
Evolocumab	1	0.4

**Table 14 biomedicines-12-02650-t014:** The Top Ten Drugs Involved in PF.

ATC Code	Drug	Description	N (Total—4452)	%
L01FA01	Rituximab	Chimeric murine/human monoclonal IgG1 antibody against CD20	245	5.5%
L01BA01	Methotrexate	4-amino-10-methylfolic acid, folic acid analog and antagonist	194	4.4%
L04AB01	Etanercept	Soluble receptor binding both TNF-alpha and TNF-beta	185	4.2%
L04AK01	Leflunomide	Monocarboxylic acid amide	178	4.0%
L04AB04	Adalimumab	fully human, recombinant Monoclonal antibody against TNF-α	164	3.7%
L04AC07	Tocilizumab	Recombinant humanized monoclonal antibody against IL-6 receptor	148	3.3%
L04AA24	Abatacept	Fusion protein that consists of the Fc portion of IgG1 combined with the extracellular domain of CTLA-4	132	3.0%
M05BA04	Alendronic acid	Nitrogen-containing bisphosphonate inhibiting bone resorption by osteoclasts	122	2.7%
L04AC10	Secukinumab	Fully human monoclonal IgG1/κ antibody against IL-17A	116	2.6%
L04AB02	Infliximab	Chimeric murine/human monoclonal IgG antibody against TNF-α	107	2.4%

Ig—immunoglobulin; IL—interleukin; CTLA-4—cytotoxic T-Lymphocyte-associated protein 4; TNF—tumor necrosis factor.

## Data Availability

Restrictions apply to the availability of these data. Data were obtained from Roszdravnadzor and are available on http://external.roszdravnadzor.ru/?type=logon with the permission of Roszdravnadzor.

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
