# Peer review of "Drug-Induced Pulmonary Fibrosis: National Database Analysis"

_biomedicines, 2024, doi:10.3390/biomedicines12122650_

Round 1

Reviewer 1 Report

Comments and Suggestions for Authors

The article “Drug-induced pulmonary fibrosis: National database analysis” by Butranova is interesting. The authors analyzed the Russian pharmacovigilance database to identify the drugs responsible for pulmonary fibrosis. However, certain corrections are required to make the article publishable.

1.     The authors included retrospective data from 2019 to mid-2024 which is around 5.5 years and found the drugs which drug is responsible for the induction of maximum PF. Whether the authors have considered the points, if any drug in this period gives less response, may induce severe PF at longer time points.

2.     Similarly, it is also not clear how many patients were taking one drug and how many had shown the PF from that group in that period. For instance, how many patients were taking busulfan while the author has reported 2 patients (0.1%). As it is only reported ADR cases in certain time points then it will be difficult to calculate the % of incidence by the particular drug in equal time points.    

3.     Is this pharmacovigilance database available online to see? Please attach a URL link following the text and table.    

4.     Please give a Y-axis outline in the figures and remove the lines in between the figures.

5.     Figure 6 needs to be visible clearly either by broken into two parts or changing its format.

6.     However, the authors have focused on PF, but the discussion part sometimes talks about the DILD which is confusing such as the discussion lines, 386-389.   

7.     Conclusion should be elaborated mechanistically. 

Author Response

Comment 1:

The authors included retrospective data from 2019 to mid-2024 which is around 5.5 years and found the drugs which drug is responsible for the induction of maximum PF. Whether the authors have considered the points, if any drug in this period gives less response, may induce severe PF at longer time points

Response to comment 1:

Thank you for comment! We have analyzed pharmacovigilance database to reveal spontaneous reports on PF entered in different years. Spontaneous reports finally analyzed were from different patients, so data for each year were unique, we did not include duplicates in our study. The increased number of spontaneous reports in 2023 and 2024 years reflected increased reporting rates for these years only, spontaneous reports on PF gained in the previous years were excluded. If the report entered pharmacovigilance database in 2019 year, it may be the result of a suspected drug use during a long time before, and only the first report was included in the analysis.  In case the same patient was identified in other report in database later it was considered as a repeat and was excluded (though there were only a few cases fitting this situation).

Comment 2:

Similarly, it is also not clear how many patients were taking one drug and how many had shown the PF from that group in that period. For instance, how many patients were taking busulfan while the author has reported 2 patients (0.1%). As it is only reported ADR cases in certain time points then it will be difficult to calculate the % of incidence by the particular drug in equal time points.

Response for comment 2:

Thank you for comment! Our work was aimed only on the assessment of the national pharmacovigilance database, which allowed us to identify the list of culprit drugs but not the size of the total sample of population used a specific drug, so incidence cannot be calculated. This is limitation of our study, and we added this section in the elaborated version of manuscript.

Comment 3:

Is this pharmacovigilance database available online to see? Please attach a URL link following the text and table.

Response to comment 3:

Thank you for comment!  Here is a link for the site of Roszdravnadzor pharmacovigilance database: http://external.roszdravnadzor.ru

But access is only for authorized users, with all information indicated on the site of National Pharmacovigilance of Russia https://roszdravnadzor.gov.ru/

Comment 4:

Please give a Y-axis outline in the figures and remove the lines in between the figures.

Response to comment 4:

Thank you for recommendation, following it we have changed figures in the manuscript.

Comment 5:

Figure 6 needs to be visible clearly either by broken into two parts or changing its format.

Response to comment 5:

Thank you for recommendation, we have changed Fig 6 to improve visualization.

Comment 6:

However, the authors have focused on PF, but the discussion part sometimes talks about the DILD which is confusing such as the discussion lines, 386-389.   

Response to comment 6:

Thank you for comment! We have made changes in the text of manuscript to clear discussion and highlight relationship between interstitial lung damage and PF development.

Comment 7:

Conclusion should be elaborated mechanistically. 

Response to comment 7:

Thank you for recommendation, following it we have changed Conclusion section in the manuscript.

Reviewer 2 Report

Comments and Suggestions for Authors

The study investigates drug-induced pulmonary fibrosis (DIPF) using a retrospective analysis of spontaneous reports (SRs) from the Russian National Pharmacovigilance database, spanning data collected from January 2019 to May 2024. This analysis seeks to identify drugs associated with higher incidences of DIPF and provides a breakdown of the implicated drug classes. Key findings emphasize antineoplastic and immunomodulating agents as significant contributors, with specific drugs such as Rituximab, Methotrexate, and Etanercept frequently reported. Additionally, the study highlights an upward trend in DIPF case reporting, potentially reflecting heightened awareness or more rigorous reporting practices in recent years.

 Major Concern:

 A key issue in this analysis is the absence of a comparison between pulmonary fibrosis cases and the overall usage frequency of these drugs in the general population. Currently, the data only identify drugs associated with pulmonary fibrosis, without indicating the proportion of patients who used these drugs but did not develop pulmonary fibrosis. This limits our ability to determine whether the increased reporting of DIPF cases is due to actual side effects or merely reflects a larger patient base using these drugs. I recommend the authors address this limitation by incorporating data on general drug usage rates or estimating relative risk more precisely. If such data are unavailable, it would be helpful for the authors to explicitly discuss this limitation in the manuscript.

 Minor Concerns:

 Given that SR-based studies inherently lack control groups, the manuscript should elaborate on the limitations of drawing causal inferences, as the absence of population-level control data constrains the robustness of conclusions.

 The study notes an increase in reported cases over recent years and attributes this trend to greater reporting awareness, which is also a key conclusion summarized in this manuscript. However, this conclusion may not be relevant or important without exploring additional factors that could impact reporting rates. It would be beneficial to consider other potential influences, such as changes in diagnostic criteria or increased usage of specific drug classes.

Comments on the Quality of English Language

There are several grammatical issues in the manuscript, which make it difficult to read. For instance, the second sentence of the abstract, The objective of our study was to reveal the structure of drugs involved in PF development analyzing national pharmacovigilance database, lacks by between development and analyzing. The authors should carefully review the manuscript for language accuracy.

Author Response

Comment 1:

Major Concern:

 A key issue in this analysis is the absence of a comparison between pulmonary fibrosis cases and the overall usage frequency of these drugs in the general population. Currently, the data only identify drugs associated with pulmonary fibrosis, without indicating the proportion of patients who used these drugs but did not develop pulmonary fibrosis. This limits our ability to determine whether the increased reporting of DIPF cases is due to actual side effects or merely reflects a larger patient base using these drugs. I recommend the authors address this limitation by incorporating data on general drug usage rates or estimating relative risk more precisely. If such data are unavailable, it would be helpful for the authors to explicitly discuss this limitation in the manuscript.

Response to Comment 1:

Thank you for comment! Analysis of the pharmacovogilance database doesn't allow to estimate the proportion of patients used drugs in the general population. This is an important limitation of our study, and we added this section in the elaborated version of manuscript.

Comment 2:

Minor Concerns:

 Given that SR-based studies inherently lack control groups, the manuscript should elaborate on the limitations of drawing causal inferences, as the absence of population-level control data constrains the robustness of conclusions. 

Response to Comment 2:

Thank you for comment! Following your recommendations we have widened Discussion section and added limitations of our study.

Comment 3:

The study notes an increase in reported cases over recent years and attributes this trend to greater reporting awareness, which is also a key conclusion summarized in this manuscript. However, this conclusion may not be relevant or important without exploring additional factors that could impact reporting rates. It would be beneficial to consider other potential influences, such as changes in diagnostic criteria or increased usage of specific drug classes.

Response to Comment 3:

Thank you for comment! We have marked in Discussion section other potential factors which may affect the rates of drug-induced PF.

Comment 4:

There are several grammatical issues in the manuscript, which make it difficult to read. For instance, the second sentence of the abstract, The objective of our study was to reveal the structure of drugs involved in PF development analyzing national pharmacovigilance database, lacks by between development and analyzing. The authors should carefully review the manuscript for language accuracy.

Response to Comment 4:

Thank you for checking grammatical issues, the final version of manuscript was elaborated with assistance of native speacker.

Reviewer 3 Report

Comments and Suggestions for Authors

The authors explored the structure of drugs associated with DIPF and their characteristics based on the national database. The list of drugs provided is comprehensive, and the statistical analyses are detailed. However, the authors should pay closer attention to the clarity of detail. Therefore, the reviewer recommended a minor revision. Key concerns are as follows:

1.     Figure 4: Are Trend lines from 1 to 10 meant to represent trends for categories from 1 to 10? If so, please clarify by using distinct or more easily distinguishable colors, as the current dashed lines are difficult to differentiate.

2.     Page 23: The author mentioned, “Formation of reactive oxygen species (ROS), DNA damage, and inhibition of new DNA synthesis may be the most common explanation.” Given the interrelation between ROS formation, DNA damage, and the inhibition of DNA synthesis, please provide robust references to support this statement.

3.     Page 24: In discussing the impact of Adalimumab on lung interstitial tissue, the authors note that “increased activity of inflammatory cells” may be a contributing factor. However, inflammatory cells such as macrophages can exhibit either pro-inflammatory (M1) or anti-inflammatory (M2) phenotypes, each associated with different inflammatory responses. Please specify which type of inflammatory cells are implicated, or use more precise terminology to clarify this point.

Author Response

Comment 1:

 Figure 4: Are Trend lines from 1 to 10 meant to represent trends for categories from 1 to 10? If so, please clarify by using distinct or more easily distinguishable colors, as the current dashed lines are difficult to differentiate.

Response to Comment 1:

Thank you for comment! In order to improve visualization we have decided to change Fig 4 and deleted trend lines.

Comment 2:

Page 23: The author mentioned, “Formation of reactive oxygen species (ROS), DNA damage, and inhibition of new DNA synthesis may be the most common explanation.” Given the interrelation between ROS formation, DNA damage, and the inhibition of DNA synthesis, please provide robust references to support this statement.

Response to Comment 2:

Thank you for comment! We have changed text to clarify this sentence ("Formation of reactive oxygen species (ROS), DNA damage and inhibition of new DNA synthesis may result in a direct cell injury which may finally lead to fibrosis [25]. ") Reference [25] is the work Li L, Mok H, Jhaveri P, Bonnen MD, Sikora AG, Eissa NT, Komaki RU, Ghebre YT. Anticancer therapy and lung injury: molecular mechanisms. Expert Rev Anticancer Ther. 2018 Oct;18(10):1041-1057. doi: 10.1080/14737140.2018.1500180

Comment 3:

Page 24: In discussing the impact of Adalimumab on lung interstitial tissue, the authors note that “increased activity of inflammatory cells” may be a contributing factor. However, inflammatory cells such as macrophages can exhibit either pro-inflammatory (M1) or anti-inflammatory (M2) phenotypes, each associated with different inflammatory responses. Please specify which type of inflammatory cells are implicated, or use more precise terminology to clarify this point.

Response to Comment 3:

Thank you for comment! 

We did use reference [Aqsa A, Sharma D, Chalhoub M. Adalimumab induced interstitial lung disease. Respir Med Case Rep. 2020 Feb 1;29:101012. doi: 10.1016/j.rmcr.2020.101012]. Authors wrote in their publication next text “The exact mechanism of pulmonary toxicity, however, remains unclear. Inhibition of inflammatory cells by anti-TNF drugs leads to unopposed activity of inflammatory cells resulting in characteristic changes of interstitial pneumonitis.”

We have added in our manuscript additional explanation linking this data with PF development:

"Mechanisms explaining lung interstitial tissue changes induced by adalimumab may include unopposed activity of inflammatory cells resulting in formation of interstitial pneumonitis [48]. This mechanism is important since the standard hypothesis describing PF pathogenesis is based on the activation of various populations of macrophages and lymphocytes resulting in cytokines and chemokines production enhancement which leads to the activation of fibroblasts and myofibroblasts and subsequent induction of transforming growth factor beta 1 (TGFβ1) and accumulation of abnormal extracellular matrix [50]."

Round 2

Reviewer 1 Report

Comments and Suggestions for Authors

The authors have responded to my quarry, however, I still see the figure's x-axis and Y-axis need to be clearly defined and lined.

I don't have further comments.

Author Response

Comment 1:

The authors have responded to my quarry, however, I still see the figure's x-axis and Y-axis need to be clearly defined and lined.

Response to comment 1:

Thank you for the work done to improve the quality of our manuscript. Figures in the manuscript are diagrams made automatically in the MS Excel. We have marked X and Y lines, they are visible as far as it possible. New version of manuscript has been uploaded.

Reviewer 2 Report

Comments and Suggestions for Authors

I have no further questions.

Author Response

Comment 1:

I have no further questions.

Response to comment 1:

Thank you very much for the work done to improve quality of our manuscript!